# A randomized multiplex CRISPRi-Seq approach for the identification of critical combinations of genes

Nicole A Ellis[1], Kevin S Myers[2,3], Jessica Tung[1], Anne Davidson Ward[1], Kathryn Johnston[1], Katherine E Bonnington[1], Timothy J Donohue[2,3,4], Matthias P Machner[1]*

[1]Eunice Kennedy Shriver National Institute of Child Health and Human Development, National Institutes of Health, Bethesda, United States; [2]Great Lakes Bioenergy Research Center, University of Wisconsin-Madison, Madison, United States; [3]Wisconsin Energy Institute, University of Wisconsin-Madison, Madison, United States; [4]Department of Bacteriology, University of Wisconsin-Madison, Madison, United States

*For correspondence: machnerm@nih.gov

Competing interest: The authors declare that no competing interests exist.

**Abstract** Identifying virulence-critical genes from pathogens is often limited by functional redundancy. To rapidly interrogate the contributions of combinations of genes to a biological outcome, we have developed a multiplex, randomized CRISPR interference sequencing (MuRCiS) approach. At its center is a new method for the randomized self-assembly of CRISPR arrays from synthetic oligonucleotide pairs. When paired with PacBio long-read sequencing, MuRCiS allowed for near-comprehensive interrogation of all pairwise combinations of a group of 44 *Legionella pneumophila* virulence genes encoding highly conserved transmembrane proteins for their role in pathogenesis. Both amoeba and human macrophages were challenged with *L. pneumophila* bearing the pooled CRISPR array libraries, leading to the identification of several new virulence-critical combinations of genes. *lpg2888* and *lpg3000* were particularly fascinating for their apparent redundant functions during *L. pneumophila* human macrophage infection, while *lpg3000* alone was essential for *L. pneumophila* virulence in the amoeban host *Acanthamoeba castellanii*. Thus, MuRCiS provides a method for rapid genetic examination of even large groups of redundant genes, setting the stage for application of this technology to a variety of biological contexts and organisms.

## eLife assessment

This **important** study uses CRISPRi to silence multiple effectors in the pathogen, *Legionella pneumophila*. It provides a technique that will allow researchers to address functional redundancy amongst effectors, a problem that has persisted even after decades of study. The methodology used is **convincing**, and further improvement (such as using multiple guides per gene) can lead to the identification of novel virulence factors.

## Introduction

There are many examples of synergistic processes in biology, often carried out by groups of redundant proteins that perform similar functions or whose activities result in the same biological outcome (*Nowak et al., 1997*; *Ghosh and O'Connor, 2017*; *Louca et al., 2018*). For example, the *Pseudomonas aeruginosa* genome encodes a total of 40 proteins for cyclic di-GMP synthesis and hydrolysis (*Ryan et al., 2006*), *Legionella pneumophila* encodes 22 (*Levi et al., 2011*), and *Vibrio cholerae*

encodes at least 11 (*Römling et al., 2013*). Genes encoding proteins of similar function are commonly acquired either through gene duplication events or horizontal gene transfer and may have been selected for to provide fail safes to vital processes. While these redundancies generally provide benefits to the organism, they are obstacles toward gaining a fundamental understanding of biological processes. In these scenarios, the disruption of a single gene does not produce a detectable growth defect, hindering further biochemical and molecular analyses.

To dissect biological processes with intrinsic redundancy, genetic approaches must be multiplexed, allowing more than one gene to be disrupted at a time. In less tractable organisms, performing traditional deletion of individual genes, let alone combinations of genes, is laborious and time consuming. Newer genetic technologies, such as clustered regularly interspaced short palindromic repeats (CRISPR)-Cas technologies allow for fast, targeted, and multiplexed gene silencing or disruption (*McCarty et al., 2020*; *Adiego-Pérez et al., 2019*). Developed from naturally occurring bacterial adaptive immune systems, CRISPR-Cas technologies are generally composed of a protein or group of proteins with enzymatic activity, usually nuclease activity, and CRISPR RNAs (crRNAs) yielding target gene specificity via homologous base pairing (*Gasiunas et al., 2012*; *Hille et al., 2018*). *Streptococcus pyogenes* Cas9 is the most commonly used nuclease enzyme that, when guided by a crRNA to a complementary target DNA location, introduces a double-strand break (*Jinek et al., 2012*; *Adli, 2018*). A catalytically inactive version of Cas9, dCas9, still localizes to complementary target genes upon crRNA direction, but instead of inducing a double-strand break, precludes RNA polymerase activity, effectively silencing gene expression. This gene silencing approach, referred to as CRISPR interference (CRISPRi), is most effective upon binding the 5' region of the gene and is often used when a biological system lacks the machinery to repair double-strand breaks produced by the original Cas9 – as is true for most bacterial organisms (*Qi et al., 2013*; *Vigouroux and Bikard, 2020*).

While silencing individual genes by CRISPRi has been achieved in various bacterial species (*Peters et al., 2016*; *Peters et al., 2019*), suppressing two or more genes simultaneously by CRISPRi has remained a major challenge as it requires the expression of multiple gene-specific crRNAs. In nature, crRNAs are encoded by spacers separated by identical repeats in long stretches of DNA known as CRISPR arrays. Customizing CRISPR arrays in the laboratory has been a daunting task as repeat-containing DNA elements are often refractory to in vitro synthesis and cloning. Recently, we succeeded in building a multiplex CRISPRi platform in *L. pneumophila* that uses synthetic arrays capable of silencing up to ten bacterial genes at a time (*Ellis et al., 2021*). Despite much progress, a major shortcoming of this and other existing multiplex CRISPRi approaches is their dependency on synthesis pipelines creating arrays of a defined set of crRNA-encoding spacers that target a predetermined set of genes (*Liao et al., 2019*; *Cress et al., 2015*; *Hawkins et al., 2015*; *Vad-Nielsen et al., 2016*; *Deaner et al., 2018*; *Zuckermann et al., 2018*; *McCarty et al., 2019*; *Shaw et al., 2022*). In cases where predictions about the number and function of genes involved in a biological process are absent or incomplete, simultaneous gene silencing by CRISPRi will fail to detect synthetic lethal combinations of genes as the redundant genes not targeted by the array remain functional. This emphasizes the need for a protocol for randomized assembly of crRNA-encoding spacers into expansive libraries of diverse CRISPR arrays for unbiased probing of varied combinations of genes. Here, we develop a <u>mu</u>ltiplex, <u>r</u>andomized <u>C</u>RISPR <u>i</u>nterference <u>s</u>equencing (MuRCiS) approach that, when used in a proof-of-concept experiment, discovered synthetic lethal combinations of genes from the virulence factor arsenal of *L. pneumophila,* the causative agent of Legionnaires' pneumonia (*McDade et al., 1977*; *Fraser et al., 1977*).

## Results
### Predetermined CRISPR arrays fail to detect virulence-critical gene combinations

*L. pneumophila* is a Gram-negative bacterium that encodes over 300 predicted virulence factors (*Burstein et al., 2016*; *Gomez-Valero et al., 2019*), known as effectors. The effectors are translocated via the Dot/Icm Type IV secretion system into either the natural host, free-living amoeba, or the disease host, human alveolar macrophages, to manipulate cellular processes and establish an intracellular replication compartment known as the *Legionella*-containing vacuole (LCV) (*Ensminger and Isberg, 2009*; *Mondino et al., 2020*; *Chauhan and Shames, 2021*). The study of *L. pneumophila*

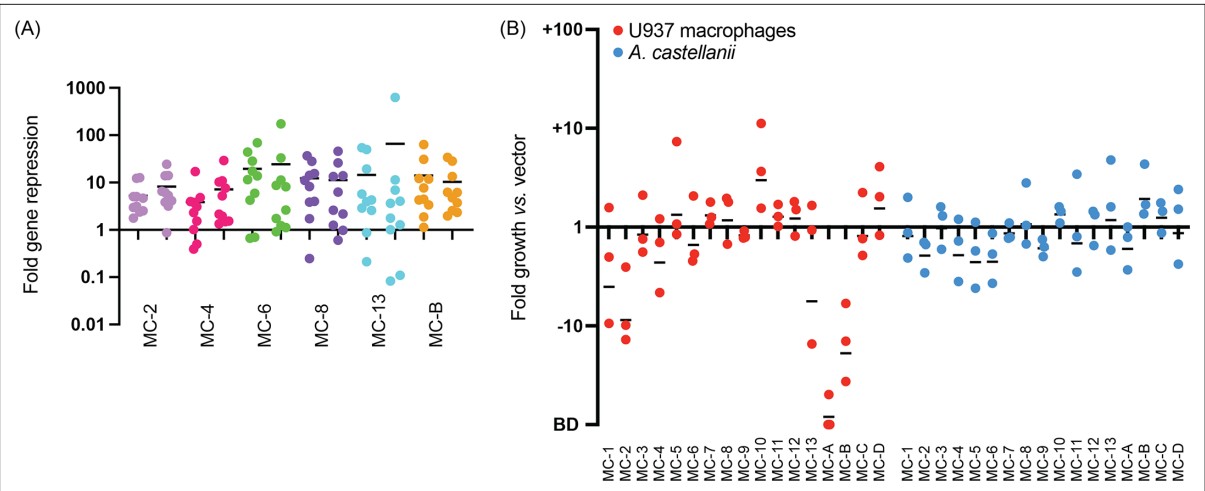

**Figure 1.** Intracellular growth of Lp02(*dcas9*) upon CRISPRi gene silencing with predetermined 10-plex arrays. (**A**) Gene silencing efficiency assay. Each of the ten genes targeted by one of the six indicated MC arrays was assayed for gene repression by quantitative polymerase chain reaction (qPCR). RNA levels in the Lp02(*dcas9*) strain bearing the CRISPR array were compared to that bearing an empty vector. Data are a summary of two replicates shown side-by-side. Horizontal bars indicate mean fold repression of all targets of the array. (**B**) *L. pneumophila* intracellular growth assay. Host cells were challenged at a multiplicity of infection (MOI) of 0.05 (U937 macrophages) or 0.03 (*Acanthamoeba castellanii*) with Lp02(*dcas9*) bearing the indicated MC arrays. Colony forming units (CFUs) of samples taken 2 hr post infection (hpi) and either 72 (U937 macrophages) or 48 (*A. castellanii*) hpi were determined, and fold growth of Lp02(*dcas9*) bearing the indicated MC arrays relative to that of Lp02(*dcas9*) bearing the empty vector was plotted. Numbers below 1 indicate a growth defect upon gene silencing. BD, below detection (arbitrarily set to –100). Horizontal bars indicate mean fold growth *vs.* vector using data from three replicates.

pathogenesis has been hindered by both the sheer number of effectors amassed by this pathogen (**Burstein et al., 2016**; **Gomez-Valero et al., 2019**) and the apparent redundancy among them, as illustrated in the lack of detectable growth defects upon disruption of individual or even entire chromosome islands of effector-encoding genes (**O'Connor et al., 2011**; **Shames et al., 2017**).

Since our earlier studies had shown that silencing of up to ten genes by multiplex CRISPRi is achievable in *L. pneumophila* (**Ellis et al., 2021**), we built a library of 10-plex CRISPRi constructs (**Supplementary file 1a**, MC, <u>m</u>ultiplex <u>C</u>RISPRi) capable of silencing more than 150 effector-encoding genes in groups of ten. The grouping of crRNA-encoding spacers into synthetic CRISPR arrays occurred based on two criteria: predicted protein function (groups 1–13, **Supplementary file 1a**) and evolutionary conservation (groups A–D, **Supplementary file 1a**). While evolutionary conservation was determined based on genome sequencing data from more than 38 *Legionella* species (**Burstein et al., 2016**), protein functions were determined based on in silico predictions using hidden Markov model (HMM)-HMM comparisons (HHPred; MPI Bioinformatics Toolkit) (**Zimmermann et al., 2018**) and transmembrane predictors (TMHMM) (**Krogh et al., 2001**). The rational was that the more conserved a gene is, the more important its biological role will be and the higher the likelihood that its silencing, either alone or in combination, will attenuate intracellular growth. Likewise, proteins of similar function, such as two effectors with kinase activity, are likely to be redundant due to a partial overlap in their range of host targets. To confirm the efficacy of multiplex gene silencing of these groups of genes, we used quantitative polymerase chain reaction (qPCR) to assess knockdown efficiency of gene expression by six of the 17 MC constructs in *L. pneumophila* bearing a chromosomal insertion of *S. pyogenes dcas9* at the *thyA* locus (Lp02(*dcas9*)) (**Ellis et al., 2021**). We found that 10-plex gene silencing was reproducible for each construct and nearly all genes were knocked down at least twofold, with an average fold repression of one order of magnitude or more (**Figure 1A**).

Upon performing intracellular growth assays with this Lp02(*dcas9*) MC strain library in both U937 human macrophages and the amoeba *A. castellanii*, we were surprised to find only a few intracellular growth phenotypes (**Figure 1B**). In U937 macrophages, the strains containing the MC-A array caused a growth defect, which was expected since it encoded crRNAs against *lpg2815* (MavN) and *lpg2300*, core effectors known to be vital for intracellular growth (**Burstein et al., 2016**). MC-10 improved growth of Lp02(*dcas9*) in U937 cells, likely due to the silencing of several glucosyltransferase effectors, *lpg1368* (Lgt1), *lpg2862* (Lgt2), and *lpg1488* (Lgt3), previously shown to act in a redundant fashion

(*Belyi et al., 2008*). MC-B caused a detectable growth defect of Lp02(*dcas9*) in U937 macrophages, suggesting it had silenced one or more virulence-critical effectors that had not been described before. Surprisingly, growth of the Lp02(*dcas9*) MC strain library in *A. castellanii* yielded no notable phenotypes. Thus, the hypothesis that *L. pneumophila* pathogenesis would be disrupted upon silencing groups of genes encoding similar or highly conserved proteins was an oversimplification, and that redundancy reached beyond the boundary of conservation or like-function. In total, these results show that silencing a predetermined set of even ten effector-encoding genes does not guarantee detection of synthetic lethal combinations of effectors.

## Self-assembly of a randomized multiplex CRISPR array library

To screen for synthetic lethality more comprehensively, we developed a protocol where CRISPR arrays were assembled de novo from oligonucleotide pairs containing crRNA-encoding spacers. Since each spacer is flanked by 36 base pair (bp) repeats of identical sequence in a canonical CRISPR array, we rationalized that these repeat sequences could be split and then re-linked during CRISPR array assembly. As such, we designed complementary DNA oligonucleotides, which we call R-S-Rs (repeat-spacer-repeat), composed of a 24 bp spacer flanked by the terminal 12 bps of the upstream repeat and the starting 20 bps of the downstream repeat, each with 4 bp 'sticky' overhangs (TGAA = a common Golden Gate cloning overhang, *Figure 2A*). The coming together of different R-S-R building blocks would recreate complete repeats without spurious nucleotides (12+20+4=classic 36 bp repeat). Since the spacer sequence was buried within the repeat sequences, competitive advantage of one R-S-R over another for integration into an array should be negligible. Furthermore, since the 4 bp overhangs are identical for each R-S-R, spacers can recombine into arrays of any length and any order. Precursor-crRNA cleavage sites were not necessary to be incorporated into the R-S-R building blocks as the precursor-crRNA would be processed into individual crRNAs by RNases endogenous to *L. pneumophila*.

As targets, we chose a total of 44 genes predicted to encode *L. pneumophila* effectors with single or multiple transmembrane domains that were identified in the above-mentioned in silico prediction. These 44 transmembrane effectors (TMEs) are likely incorporated either into the membrane of the LCV or that of surrounding organelles to regulate membrane dynamics or the transport of metabolites (nutrients or waste products) across the membrane (*Figure 2—figure supplement 1*, *Supplementary file 1b*). Such TMEs, when absent due to gene silencing, are unlikely to be replaced by cytosolic effectors, suggesting that redundancy is more likely to be detectable within this group of TMEs. And since transmembrane regions can be predicted with high confidence, we decided to probe this group of genes for synthetic lethality with the randomized CRISPRi approach as proof-of-concept.

Assembly of CRISPR arrays followed simple, canonical cloning procedures. CRISPR arrays were allowed to self-assemble after brief heat denaturation and slow cooling of 44 complementary R-S-R oligonucleotide pairs with an aliquot of the 'dead end' oligonucleotide pairs, attB4r-R and R-attB3r (*Figure 2B*), to cap arrays on either end (*Figure 2C*). Assembled arrays were locked into place by treatment with T4 ligase and incorporated into an interim Invitrogen Gateway cloning plasmid, pDonor-P4r-P3r, by way of the attB4r/attB3r sequences.

At this point, we found it critical to size-select arrays, as shorter fragments of only one or two spacers efficiently outcompeted longer fragments for vector incorporation, a common phenomenon of basic cloning. Size selection proved challenging (Appendix 1) but was ultimately accomplished through restriction enzyme-based excision of the arrays from the interim plasmid and DNA gel purification of arrays of a desired size. To select for intermediate (2–4 spacers) and long (4+ spacers) arrays, gel fragments corresponding to 550–650 bps and 650–800 bps were purified, respectively. Purified arrays were then ligated back into the interim Invitrogen Gateway cloning plasmid.

Last, we used an Invitrogen Multisite Gateway Pro cloning strategy to create a final vector containing the tracrRNA-encoding sequence, and the pooled, size-selected randomized CRISPR array library flanked by a promoter (P$_{tet}$) and terminator (*rrnB* T1) sequence. Keeping the tracrRNA separate from crRNAs, unlike in single guide RNAs, kept the R-S-R building blocks at a manageable length. Upon navigating this assembly protocol twice, the final vector libraries were introduced into Lp02(*dcas9*) by electroporation and advanced to the next stage.

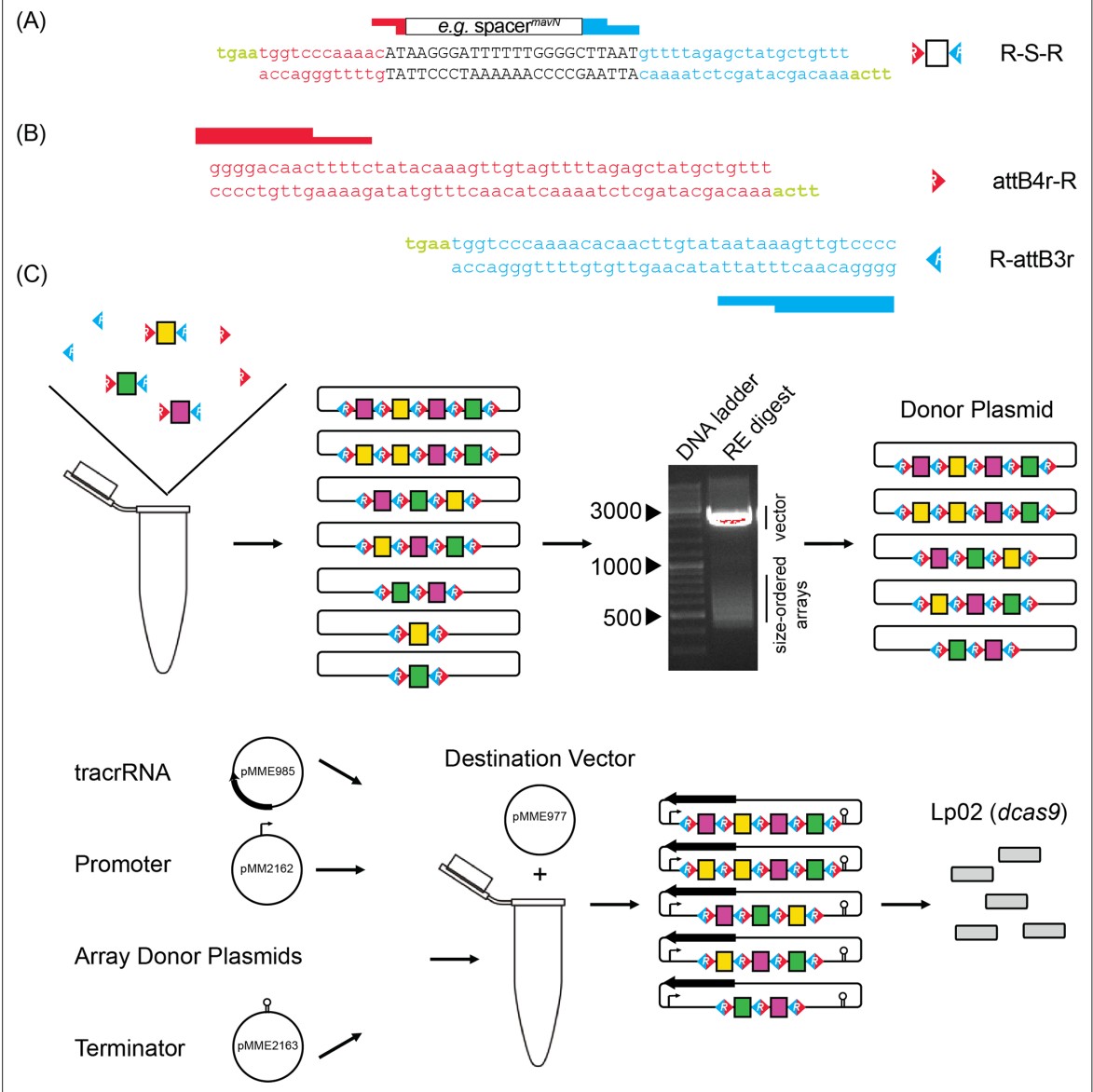

**Figure 2.** Cloning strategy for de novo self-assembly of CRISPR array libraries. (**A**) Nucleotide sequence of R-S-R building blocks. Each R-S-R element is composed of a top and bottom oligonucleotide. The 24 bp spacer sequence (black font; shown for crRNA$^{mavN}$ as an example) is flanked by sequences of the upstream (red) and downstream (blue) repeat element, with each end containing sticky overhangs (green). (**B**) Nucleotide sequence of attB4r-R and R-attB3r 'dead ends' with only one sticky overhang each (green). (**C**) Array self-assembly, size selection, and cloning. R-S-Rs and 'dead end' elements were allowed to self-assemble in a single tube and then ligated together and incorporated into an interim cloning vector. Arrays excised from an interim cloning vector were subjected to gel electrophoresis to separate them according to size. RE, restriction enzyme. After gel extraction, arrays of 550–650 bps and 650–800 bps in size were cloned into a donor plasmid such that three additional elements, namely the tracrRNA, a *tet* promoter, and a *rrnB* T1 terminator, could assemble with the arrays into the destination vector by Invitrogen Multisite Gateway Pro cloning. Final constructs were introduced into Lp02(*dcas9*) by electroporation.

The online version of this article includes the following figure supplement(s) for figure 2:

**Figure supplement 1.** Model of the *Legionella*-containing vacuole bearing transmembrane effectors.

## Self-assembled CRISPR arrays are diverse in length, order, and composition

To assess the diversity of array composition, CRISPR array-containing vector libraries (Libraries 1 and 2) were harvested from two Lp02(*dcas9*) subpopulations, linearized, and analyzed by next-generation sequencing. Importantly, given the repetitive nature of the CRISPR arrays, canonical short-read

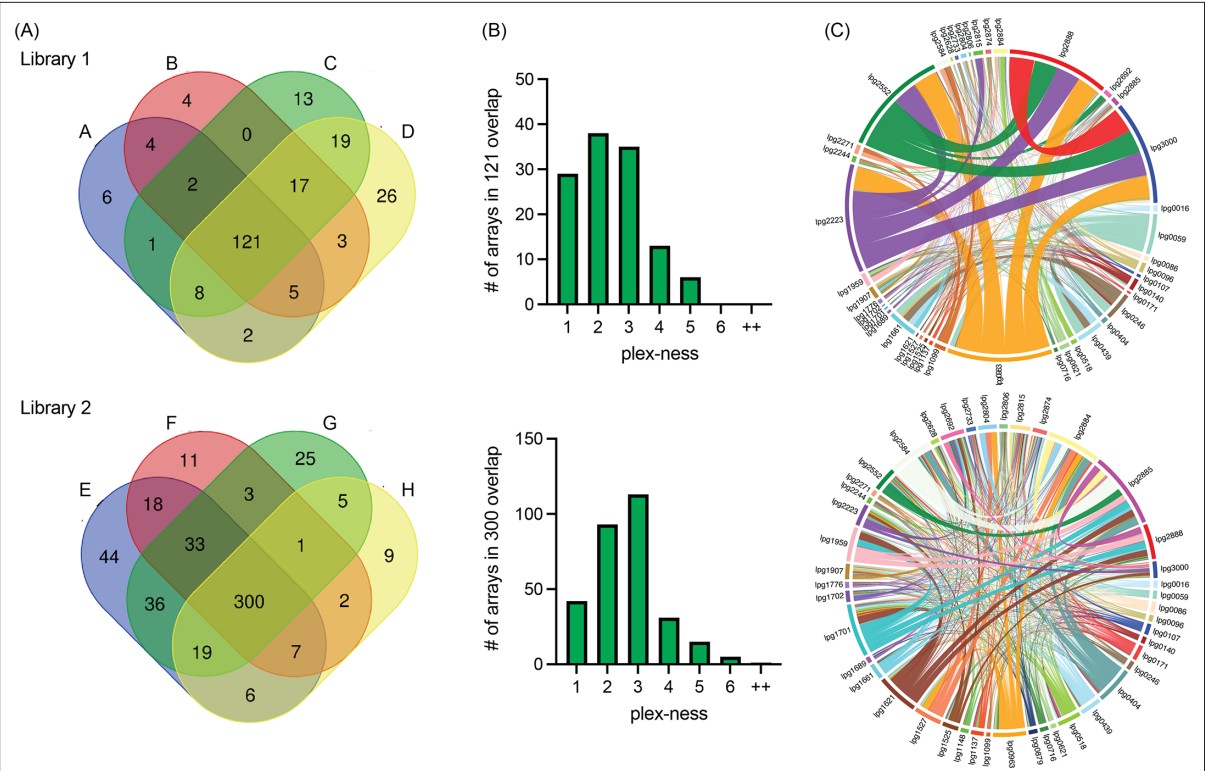

**Figure 3.** Self-assembled CRISPR arrays are diverse in length and spacer composition. (**A**) Two CRISPR array libraries were built from 44 R-S-R building blocks. Vectors from each library were harvested four times and sequenced. Venn diagrams show the overlap of unique spacer combinations, with the requirement of five or more raw (pre-sequence-depth normalized) read counts per array in each round of sequencing. (**B**) The distribution of plex-ness (spacer count) for arrays found in all four rounds of sequencing. (**C**) Chord diagrams link transmembrane effector (TME) genes, listed around the outside of the circle, each time spacers targeting each is present in an array. Link line width is weighted according to the number of times the combination of spacers was observed. Link line color is unique for each spacer and is constant between the two diagrams.

The online version of this article includes the following figure supplement(s) for figure 3:

**Figure supplement 1.** Spacer abundance post CRISPRi induction.

sequencing technologies (such as Illumina or 454 sequencing) would not have been adequate as they would not have provided enough unique sequence overlap outside of the repeat sequence to map the reads back to a specific array within the mixed vector population. Instead, we made use of a high-throughput, low error rate long-read sequencing technology (PacBio Sequel). The average length of reads containing R-S-R elements was ~10,400 bp, correlating to the size of the vector backbone plus a multiplex array (sequencing metrics reported in *Supplementary file 1c*).

We harvested vectors four times from each Library 1 and Library 2 stock to assess array diversity and library reproducibility. We designed a custom bioinformatics pipeline (discussed below) to condense data from all arrays with identical spacer content into one dataset, regardless of spacer order; as such we found that Library 1 contained 784 unique spacer combinations and Library 2 contained 1251 unique spacer combinations (overlap = 250 combinations found in both). When requiring a stringent arbitrary cut-off of five or more raw (pre-sequence-depth normalized) read counts per spacer combination, a total of 231 and 519 spacer combinations were observed in Libraries 1 and 2, respectively, with 121 and 300 of those spacer combinations being present in each of the four replicates (A–D and E–H; *Figure 3A*). We found that arrays of spacers were most often 2- to 3-plex in nature, though some reached 5-, 6-, or even up to 11-plex to give an average of 3.3 spacers per array (*Figure 3B*, *Supplementary file 1d*).

Each library contained a diversity of arrays that would assess silencing of nearly all 946 possible pairwise combinations of the 44 TME genes. Chord diagrams plotting each time spacers targeting two different TME genes were found in the same array provided a visualization of the remarkable pairwise comprehensiveness of each library, with the weight of the linking lines indicating frequency

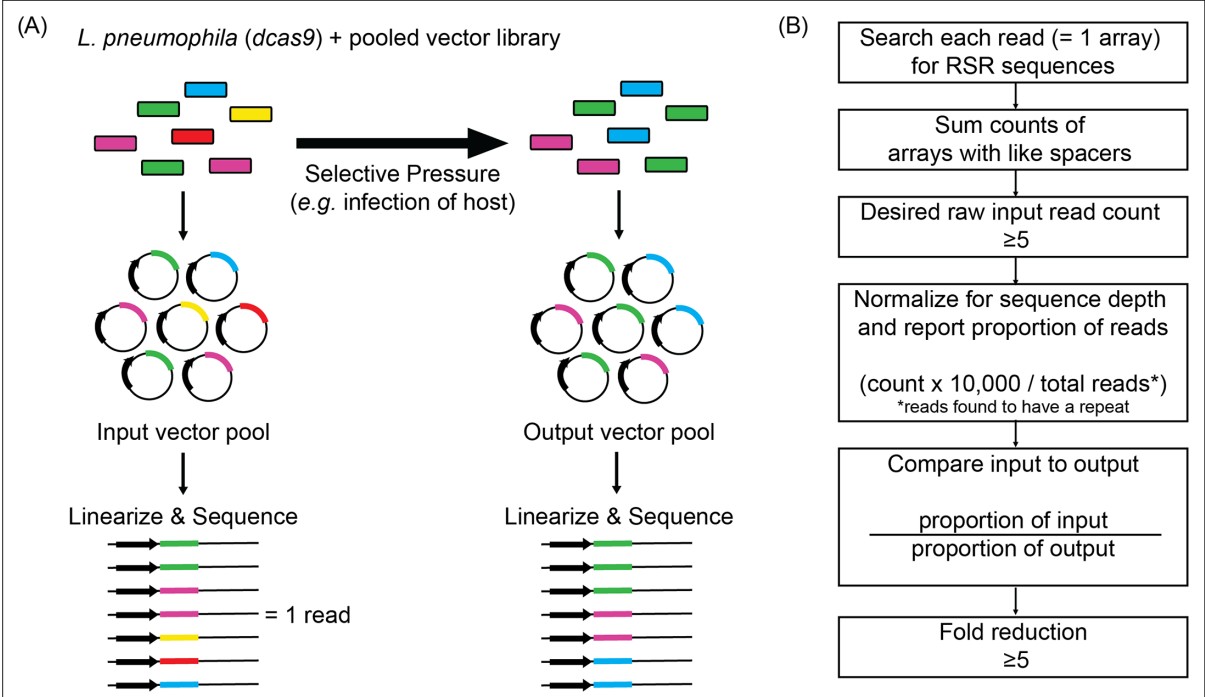

**Figure 4.** Schematic overview of the experimental and bioinformatics pipeline of MuRCiS. (**A**) During MuRCiS, a pooled population of *L. pneumophila* (*dcas9*) bearing the multiplex random CRISPR arrays were subjected to a selective pressure (intracellularity). Vectors were purified from both input and output bacteria populations, linearized, and submitted for long-read PacBio Sequel sequencing. For simplification, an array of mixed spacer population is shown as a single stretch of color, each color representing a different combination of spacers. In this example, the yellow and red arrays are lost in the output suggesting they silence critical combinations of genes. (**B**) Overview of the custom bioinformatics pipeline used to identify unique spacer combinations causing growth attenuation as defined by a fold reduction of five or more (https://github.com/GLBRC/MuRCiS_pipeline,copy archived at *Myers and Donohue, 2023*).

of observation (*Figure 3C*). Both libraries showed vast diversity in array composition, with Library 2 showing the greatest randomization of arrays leading to the most comprehensive pairwise coverage. Library 1, surprisingly, had an over-representation of arrays targeting the same five genes (*lpg0963*, *lpg2223*, *lpg2552*, *lpg2888*, and *lpg3000*) that accounted for 20% of all arrays in that library (indicated by the thick network of linking lines). Notably, from an array building vantage, the 5-plex arrays targeting these five genes were found to have seven different spacer orders, indicating that they had been randomly generated multiple times during the assembly step.

Last, we examined the overall abundance of each spacer in the libraries to verify that no one crRNA-encoding spacer was individually toxic, that is, targeting a gene essential for *L. pneumophila* axenic growth (intended or off-target). When combining data from all libraries, each spacer was present a median of ~5328 times (*Figure 3—figure supplement 1*). No one spacer was under-represented, confirming none was toxic, but also that each R-S-R element had an equal opportunity to be incorporated in an array. Together, these two libraries provide a vast collection of multiplex arrays to test for combinatorial TME gene contributions to *L. pneumophila* pathogenesis strategies.

## Decoding randomized CRISPR array libraries by high-throughput sequencing

To identify combinations of genes that, upon silencing, were detrimental to intracellular *L. pneumophila* replication, vectors from Lp02(*dcas9*) cultures bearing the pooled input array libraries were purified, linearized, and PacBio Sequel-sequenced to determine the identity and composition of arrays before infection (*Figure 4A*). From the same cultures, pooled bacteria were simultaneously introduced to a selective pressure, in this case host cell infection, and surviving bacteria were harvested from host cells after 72 hr (for U937 macrophage infections) or 48 hr (*A. castellanii* infections) and grown as

single colonies on media plates. Vectors from arising bacterial colonies were purified, linearized, and analyzed again by long-read sequencing to determine the identity of arrays in the output vector pool.

Since each PacBio sequencing read reported the complete spacer composition of an array, barcoding was unnecessary. A custom read count-based bioinformatics pipeline was designed to determine arrays that were under-represented in the output pool, and therefore, must have silenced genes vital for successful intracellular replication (*Figure 4B*, https://github.com/GLBRC/MuRCiS_pipeline, copy archived at *Myers and Donohue, 2023*). Taking a most stringent approach, only reads that were an exact match to the repeat sequence were analyzed. Spacers within an array were identified by searching each read for exact matches to all possible forward and reverse combinations of spacer sequences and the repeat sequence. The number of arrays with the same spacer composition were counted and an arbitrary cut-off of five raw counts in the input pool was required for further analysis of that unique spacer combination. The remaining unique array counts were normalized for run-to-run sequence depth variation, and the proportion of each unique array compared to the total number of sequenced reads containing a repeat sequence was determined for each sample. For each experiment, the proportion of normalized counts for each unique spacer composition in the input sample was compared to the normalized counts for the same spacer composition in the output sample to determine a fold reduction. The larger the reduction, the more disadvantageous silencing of the targeted genes was for intracellular *L. pneumophila* replication. We excluded any subset of

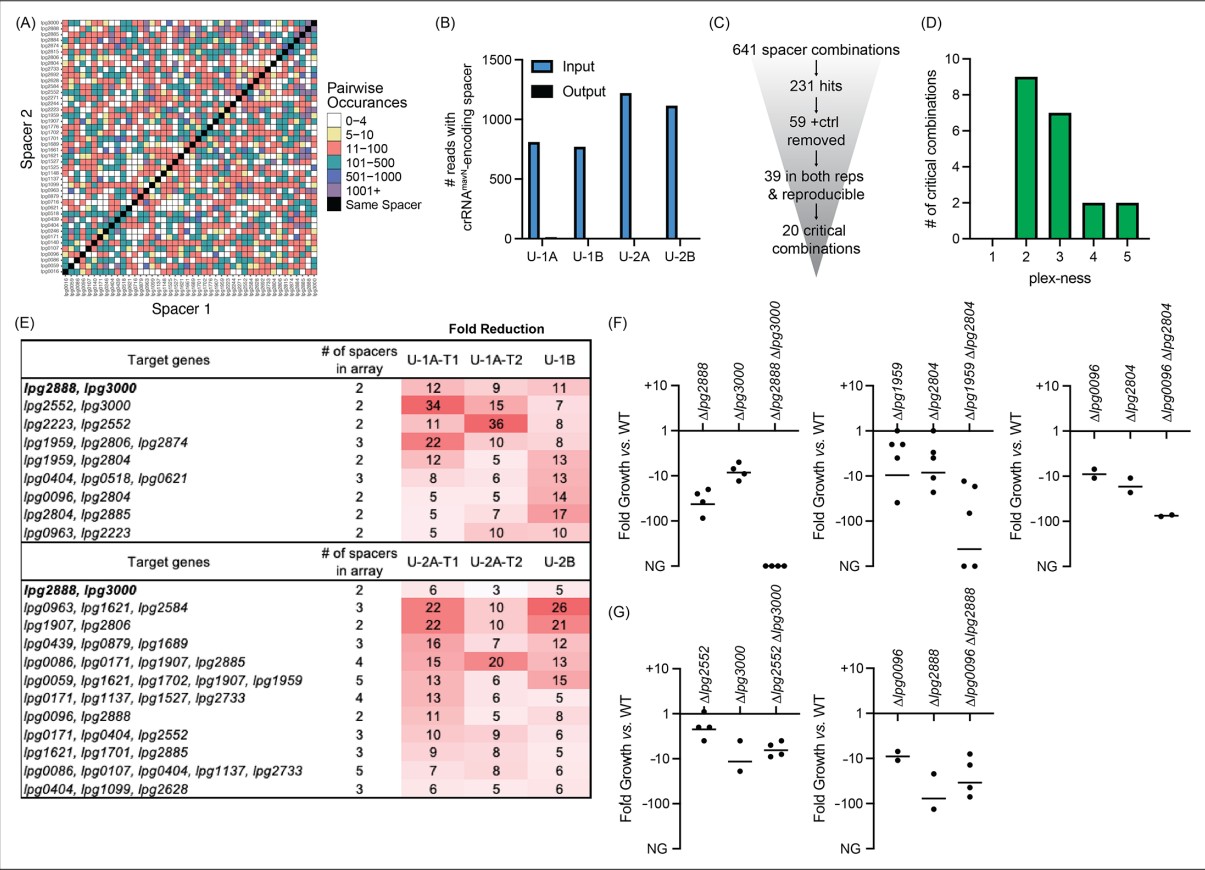

**Figure 5.** Discovery of *L. pneumophila* gene combinations critical for growth in U937 macrophages. (**A**) Correlation grid plotting all pairwise spacer combinations. The number of times two unique spacers were present in the sequenced array library is indicated by color-coding, ranging from light (rare) to dark (abundant). Black boxes, same spacer. (**B**) Total number of reads bearing the spacer which encodes crRNA$^{mavN}$. Counts in technical output replicates were summed. (**C**) Flowchart going from the number of total spacer combinations to the number of critical combinations identified. (**D**) Histogram indicating the plex-ness of all critical spacer combinations identified. (**E**) A list of virulence-critical combinations of genes and the corresponding fold reduction observed for each in each experiment. Library 1 results are shown above Library 2 results. (**F**) Intracellular growth assays of strains with deletions in the indicated genes. Results are given as fold growth (colony forming units [CFUs] harvested 72 hr post infection (hpi) *vs.* 2 hpi) compared to that of Lp02 (WT). Horizontal bars indicate mean fold growth of the deletion strain *vs.* WT strain for two or more experimental replicates. NG, no growth (arbitrarily set to –1000). (**G**) Intracellular growth assays of deletion strains that do not have a synergistic phenotype.

spacers with an input *vs.* output reduction of less than fivefold. Together, this experimental and bioinformatics pipeline forms the basis for MuRCiS.

## MuRCiS elucidates novel combinations of virulence-critical genes

We first interrogated the contributions of TME genes to successful intracellular replication of *L. pneumophila* in human U937 macrophages. We performed four rounds of infections: two with Lp02(*dcas9*) bearing Library 1 (U-1A and U-1B) and two with Lp02(*dcas9*) bearing Library 2 (U-2A and U-2B). Each pair of infections, A and B, occurred independently of each other serving as biological replicates. To test technical reproducibility, we also collected output samples twice (T1 and T2) when performing experiments 'A'. The four inputs for these experiments were A, C, E, and G (*Figure 3A*). In total, the four input cultures used to challenge U937 macrophages allowed for the probing of 710 of a possible 946 pairwise combinations (75%) of 44 TME when requiring a stringent arbitrary cut-off of five or more raw read counts (*Figure 5A*, *Supplementary file 1d*). In fact, pairs were often in the input pool between 11 and 500 times allowing for in-experiment replication. Silencing *mavN,* which encodes the essential metal ion transporter (*Isaac et al., 2015*, *Christenson et al., 2019*), served as a control for the infection experiments. While hundreds to thousands of reads bearing the crRNA$^{mavN}$-encoding spacer were present in each input pool, each of the output samples (T1/T2 summed) had only ten or fewer reads (*Figure 5B*). These results indicated that MuRCiS could identify virulence-critical genes by monitoring changes in spacer abundance during host cell infection.

In total, 641 unique spacer combinations with five or more raw input reads were tested using this protocol (*Figure 5C*). Subsequent bioinformatic analyses found that 231 of those combinations showed a fivefold or greater reduction in read counts. All 59 combinations which contained the crRNA$^{mavN}$-encoding spacer were removed from the list, and 39 spacer combinations were identified as having a fivefold or greater read count reduction in both replicates of each library experiment (including both technical replicates). These 39 spacer combinations were examined further by gathering shorter versions of the arrays to track down the minimal number of spacers needed for the strain to show a fitness defect (*Supplementary file 1e*). Twenty critical combinations of spacers were identified, of which most were 2- to 3-plex combinations (*Figure 5D and E*). Because of the uniqueness of Libraries 1 and 2, different gene combinations were discovered by each library to be virulence-critical, suggesting that the assembly and assay of more than one library was advantageous for comprehensiveness. Targeting of *lpg2888* and *lpg3000* was identified to be deleterious to *L. pneumophila* intracellular growth in experiments with both libraries. *Supplementary file 1f* shows data for each of the subsets of the 20 critical gene combinations, providing evidence that it was truly the silencing of the reported combination of genes, and not fewer or individual genes, that was needed for the strain to show a fitness defect.

## Multiple TME pairs are critical for *L. pneumophila* virulence in macrophages

The nature of CRISPRi gene silencing and the novelty of the screen performed here mandated confirmation of targets by construction of strains with true chromosomal gene deletions, as the degree of target silencing can be influenced by a range of factors related to (1) crRNA design, (2) genetic environment, and on occasion (3) off-target effects. Individual and pairwise deletions of the candidate genes were made for five of the combinations identified in *Figure 5E*. All deletion strains were whole genome sequenced for confirmation of gene deletion and absence of background mutations and then used to challenge macrophages. Their ability to survive and grow over a period of 3 days was determined by plating assay.

This analysis confirmed the expected growth-inhibitory phenotype for three of the five gene pair deletions (*Figure 5F*), namely *lpg2888-lpg3000*, *lpg1959-lpg2804*, and *lpg0096-lpg2804*. The severity of the phenotype varied among the different pairs, ranging from an ~75-fold reduction in growth to complete inability to replicate in the host. For two pairs, *lpg2552-lpg3000* and *lpg0096-lpg2888*, we found that deleting only one of the genes, *lpg3000* and *lpg2888*, respectively, was sufficient for limiting *L. pneumophila* intracellular replication (*Figure 5G*). While we aimed to find critical combinations of genes, these 'false positive hits' still identified genes important to *L. pneumophila* intracellular replication demonstrating much can be learned from this assay.

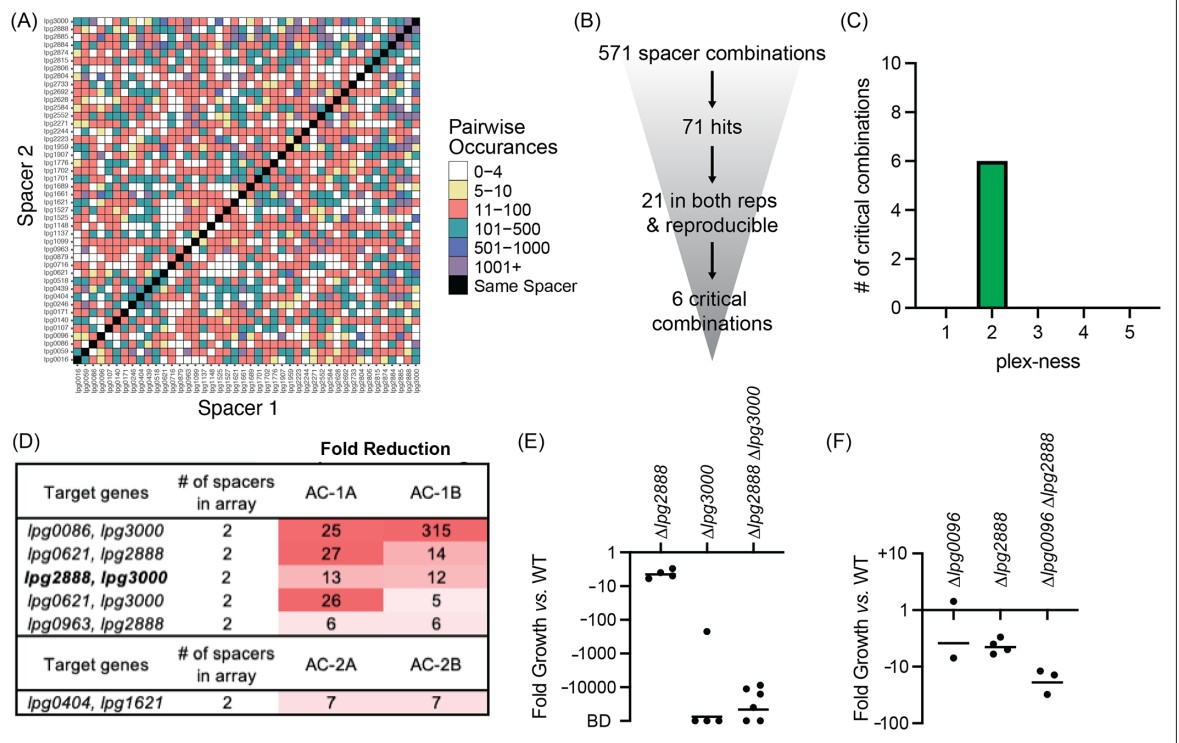

**Figure 6.** Discovery of *L. pneumophila* gene combinations critical for growth in *A. castellanii*. (**A**) Correlation grid plotting all pairwise spacer combinations. The number of times two unique spacers were present in the sequenced array library is indicated by color-coding, ranging from light (rare) to dark (abundant). Black boxes, same spacer. (**B**) Flowchart going from the number of total spacer combinations to the number of critical combinations identified. (**C**) Histogram indicating the plex-ness of all critical spacer combinations identified. (**D**) A list of virulence-critical combinations of genes and the corresponding fold reduction observed for each in each experiment. Library 1 results are shown above Library 2 results. (**E**) Intracellular growth assays of strains+pMME2400 with deletions in the indicated genes. Results are given as fold growth (colony forming units [CFUs] harvested 48 hr post infection [hpi] *vs*. 2 hpi) compared to that of Lp02+pMME2400 (WT). Horizontal bars indicate mean fold growth of the deletion strain *vs*. WT strain for two or more experimental replicates. BD, below detection (arbitrarily set to –100,000). (**F**) Intracellular growth assays of strains with deletions in the indicated genes having been identified in single-round CRISPRi experiments.

## Virulence-critical gene combinations are host-specific

Having found critical combinations of genes required for growth of *L. pneumophila* in its disease host, we next interrogated the cohort of TME genes for their necessity during *L. pneumophila* growth in the natural host, *A. castellanii*. As before, we performed four rounds of infections: two with Lp02(*dcas9*) bearing Library 1 (AC-1A and AC-1B) and two with Lp02(*dcas9*) bearing Library 2 (AC-2A and AC-2B). The four inputs for these experiments were B, D, F, and H (*Figure 3A*). In total, these four inputs allowed for probing 672 of a possible 946 pairwise combinations (71%) of 44 TME genes when requiring a stringent arbitrary cut-off of five or more raw read counts (*Figure 6A*, *Supplementary file 1d*).

This time, 571 unique spacer combinations with five or more raw input reads were tested during the experiments, leading to the identification of 71 spacer combinations with fivefold or greater reduction in read counts (*Figure 6B*). Of those, 21 emerged from both replicates of a given library experiment. We examined these spacer candidates further by tracking the most minimal number of spacers needed to attenuate *L. pneumophila* intracellular growth (*Supplementary file 1g*). Six critical combinations of TME genes were ultimately identified, of which all were 2-plex combinations (*Figure 6C and D*, *Supplementary file 1h*). Notably, silencing of *lpg2888* and *lpg3000* was again found to be deleterious to *L. pneumophila* growth in *A. castellanii*, as it was in U937 macrophages (*Figure 5E*). Having already constructed *L. pneumophila* stains bearing deletions in *lpg2888* and/or *lpg3000*, we tested these for growth in *A. castellanii*. Surprisingly, in *A. castellanii* deletion of *lpg3000* alone was sufficient to cause a growth defect as dramatic as that observed for the double mutant (*Figure 6E*). This contrasted with U937 macrophage experiments where only deletion of both *lpg2888* and *lpg3000* resulted in attenuated growth (*Figure 5F*).

Two of the other critical virulence-gene combinations identified in the *A. castellanii* experiments also contained *lpg3000*: *lpg0086-lpg3000* and *lpg0621-lpg3000,* suggesting we would again observe deletion of *lpg3000* sufficient of a maximum intracellular growth defect if tested. Though each library included 1-plex arrays expressing only crRNA$^{lpg3000}$ for silencing *lpg3000*, we did not identify those 1-plex arrays as being sufficient to attenuate virulence, likely because their decrease was not consistent between Library 1 and Library 2 experiments, although there was consistency between replicates A and B of each library (*Supplementary file 1h*). A possible explanation was that incomplete knockdown (*vs.* complete knockout) of a seemingly important process was at a tipping point of phenotype extremes, and that silencing of an additional effector may have tipped the scale. If requiring a fivefold or greater reduction in only one library and not the other, an additional three (two of which are 1-plex; *lpg0086* and *lpg2223*) and six (one of which is 1-plex; *lpg0086*) spacer combinations may qualify as critical to U937 macrophage and *A. castellanii* infection, respectively.

## Single-round CRISPRi experiments are of sufficient sensitivity to detect virulence-critical genes

For any virulence-critical gene combination to be listed as a hit during U937 macrophage (*Figure 5*) or *A. castellanii* infection (*Figure 6*), the CRISPR arrays silencing them had to pass the most stringent cut-off criteria, including that the spacer combination had to be represented in both biological replicates, A and B. There were instances though where CRISPR arrays were reduced in only one replicate because arrays bearing the same spacer combinations were absent in the other replicate (input overlap shown in *Figure 3A*). When requiring a fivefold or greater reduction in only one replicate, an additional 18 and 14 virulence-critical spacer combinations were observed in the U937 macrophage and *A. castellanii* experiments, respectively (*Supplementary file 1i*). For example, in the infection experiment AC-2A, arrays silencing both *lpg0096* and *lpg2888* were ninefold under-represented in the output pool after *L. pneumophila* growth in *A. castellanii*. *L. pneumophila* strains bearing deletions in either one or both genes were examined for growth in *A. castellanii*. Interestingly, unlike in U937 macrophages, we did observe a synergistic phenotype upon deletion of *lpg0096* and *lpg2888* (*Figure 6F*). These data represent yet another example of a host-specific phenotype and suggest that the proof-of-concept study performed here may have identified even more virulence-critical combinations of *L. pneumophila* genes than initially thought warranting examination of gene combinations that are considered hits in only one of the experimental replicates.

## Discussion

In this study, we expanded the versatility of CRISPRi platforms in bacteria by devising an approach capable of 'shuffling' the spacer composition of CRISPR arrays, thus creating a randomized, multiplexed genetic tool adept to cross-examine even large cohorts of *L. pneumophila* effectors for synthetic lethality. A critical innovation of our approach was the quantitative (abundance output vs input pool) and qualitative (spacer composition) analysis of CRISPR arrays by high-throughput long-read sequencing. Indubitably, MuRCiS will be applicable to the study of any group of redundant genes in a wide range of CRISPRi-capable organisms.

Our de novo assembly protocol created distinct libraries of arrays each time it was executed, exemplifying its truly random nature (*Figure 3*). As mentioned, Library 1 unexpectedly amassed arrays targeting five specific genes (*lpg0963*, *lpg2223*, *lpg2552*, *lpg2888*, and *lpg3000*), such that these arrays accounted for 20% of the input read counts for this library. These arrays did decrease in abundance during *L. pneumophila* infection of both U937 macrophages and *A. castellanii,* consistent with our finding that *lpg2888* and *lpg3000* are important for replication in both these hosts. We do not fully understand why these particular arrays were so prevalent only in Library 1, even more so given that assembling arrays of fewer than five spacers proved more efficient (*Figure 3B*). It is possible that silencing these TME genes provided *L. pneumophila* with a fitness advantage during axenic growth causing them to be over-represented in the input pool.

A unique feature of MuRCiS is its ability to quickly assign phenotypes to exact spacer combinations simply by tracking phenotypes seen with longer arrays to shorter arrays with subsets of spacers and, hence, target genes for subsequent follow-up studies (*Supplementary file 1e* and *Supplementary file 1g*). For example, the 5-plex arrays containing *lpg0963*, *lpg2223*, *lpg2552*, *lpg2888*, and *lpg3000*

can be subdivided into 30 subsets of spacers – all of which were present in the experiment and allowed assigning the phenotype to just two genes, *lpg2888* and *lpg3000* (*Supplementary file 1e*). Notably, not all phenotypes caused by long arrays could be condensed to just one or two spacers. Some phenotypes required 4- or 5-plex gene silencing (*Supplementary file 1f*), showing that although only a minor fraction of all possible combination of three or more spacers were represented in our array library, the approach was capable of identifying these synthetic lethal gene silencing combinations as well.

Examining the PacBio DNA sequencing data, we found that the number of unique arrays identified in a given sample increased as the read number increased (*Supplementary file 1c*). For this proof-of-concept experiment we chose to limit the initial pool of arrays for analysis to those with a minimum of five raw input counts. Maintaining this same cut-off with increased sequence depth undoubtedly would increase the input-to-input array overlay (*Figure 3A*) and the comprehensiveness of surveying pairwise silencing of genes (*Figures 5A and 6A*). We anticipated the gains of increased sequencing depth by recalculating the coverage of the correlation plots in *Figures 5A and 6A* such that the observed coverage of 710/946 (75%) or 672/946 (71%) pairs with the requirement of five raw input counts increased to 789/946 (83%) or 782/946 (83%) pairs when the requirement was lowered to just one raw input count, respectively. Hence, the ability to make comprehensive libraries was even better than first reported.

Array length poses the biggest bottleneck to performing MuRCiS on larger groups of genes. Here, each of the libraries individually probed an average of ~1000 different spacer combinations (independent of order). If all spacer combinations were represented by 2-plex arrays, theoretically one could investigate pairwise combinations of ~45 genes using just 990 arrays. But as reported, the assembly protocol produced longer arrays of up to 11 spacers, averaging 3.3 spacers. Each time the average length of arrays is increased by one spacer, the number of pairwise combinations able to be probed dramatically increases. If all arrays were 3-plex, 3000 pairwise combinations would be possible within just 1000 arrays allowing comprehensive interrogation of a group of ~75 genes for synthetic lethality, and if all arrays were 4-plex, 6000 pairwise combinations would be possible within just 1000 arrays allowing comprehensive interrogation of a group of ~110 genes for synthetic lethality. As it stands, the theoretical upper limit of MuRCiS seems between 75 and 100 genes, but with the construction of more than one library, deeper sequencing, and lowered input count requirements this number could easily be expanded.

Excitingly, the combinatorial effector gene deletions that were assayed in our experiments produce some of the largest phenotypes yet identified pertaining to *L. pneumophila* intracellular replication (*Figures 5 and 6*). This is especially true of the simultaneous deletion of *lpg2888* and *lpg3000* producing a truly synthetic lethal phenotype during U937 macrophage infection. The only other pair of *L. pneumophila* genes known to have such a large combinatorial impact on *L. pneumophila* intracellular replication are *icmW* and *icmR* (*Coers et al., 2000*), genes that encode components of the Dot/Icm Type IV secretion system itself; hence their deletion has a global effect limiting translocation of many effector proteins into the host. HHpred analysis of *lpg2888* revealed numerous high confidence hits to tripartite pore-forming toxin components, while *lpg3000* showed high confidence hits to ABC transporter proteins. Additional studies are ongoing to uncover why deletion of their encoding genes caused synthetic lethality. From an evolutionary standpoint, it was interesting that synergy of these genes was only seen during *L. pneumophila* infection of U937 macrophages. During *A. castellanii* infections, deletion of just *lpg3000* was already lethal, whereas deletion of *lpg2888* alone caused only a minor growth defect (*Figure 6E*). These data are in agreement with data from a transposon sequencing (Tn-Seq)-based screen that identified a growth defect upon disruption of *lpg3000* in *L. pneumophila* during infection of *A. castellanii* and another protist, *Hartmannella vermiformis* (*Park et al., 2020*). Altogether, these results suggest that the biological process promoted by *lpg3000* is evolutionarily essential for growth in the natural amoeban host. Clearly, MuRCiS identified exciting biology pertaining to *L. pneumophila* pathogenesis; and if applied to other groups of genes, other selective pressures, or other microorganisms, it undoubtedly has the potential to do the same.

## Materials and methods

### Strains and cell lines

All *L. pneumophila* strains are listed in *Supplementary file 1j* and were derived from the parent strain *L. pneumophila* Philadelphia-1 Lp02 (*thyA hsdR rpsL*). Host cell lines included U937 monocytes (ATCC CRL-1593.2) which were differentiated into macrophages with 12-*O*-tetradecanoylphorbol-13-acetate (TPA, Sigma-Aldrich P1585) and *A. castellanii* (ATCC 30234).

### Synthetic MC array construction

Synthetic 10-plex MC array constructs were synthesized by GenScript and moved into pMME977 (*thyA+*) by Multisite Gateway Pro cloning (Invitrogen 12537-100) as previously described (*Ellis et al., 2021*). The MC arrays tested here bear a *boxA* sequence –58 bp upstream of the first repeat which was added after vector completion by quick-change PCR as described previously (*Ellis et al., 2021*). Final plasmids were introduced to Lp02(*dcas9*) by electroporation and the strains containing the plasmids were selected for on CYE plates (*Feeley et al., 1979*) without thymidine. Spacer composition of each array and their corresponding nucleotide sequences are listed in *Supplementary file 1a*. All strains are available upon request.

### Axenic growth of *L. pneumophila* 10-plex CRISPRi strains

As described previously (*Ellis et al., 2021*), *L. pneumophila* cultures were grown overnight in AYE+Fe+Cys (10 g ACES, 10 g yeast extract per liter, pH 6.9 with 0.4 mg/mL cysteine, and 0.135 mg/mL ferric nitrate) under non-inducing conditions. On the second day, cultures were sub-cultured twice (AM and PM, ~6–7 hr apart) to OD600 0.2–0.3 with 2–3 mL fresh AYE+Fe+Cys containing 40 ng/mL anhydrous tetracycline (aTC, Clontech 631310). On the third day, cultures that had reached OD600 3–5 (post-exponential growth) were collected for mRNA analyses and/or used in host cell infections.

### RNA extraction and qPCR

RNA extraction was performed on bacteria pellets after aTC induction using the Trizol Max Bacterial RNA Isolation Kit (Invitrogen 16096040). Contaminating DNA was removed using the Turbo DNA-free Kit (Invitrogen AM1907) and RNA was converted to cDNA using the High-Capacity cDNA Reverse Transcription Kit (Applied Biosystems 4368814). qPCR was performed using the SYBR Green Master Mix (Applied Biosystems 4367659) on a StepOne Plus Real-Time PCR System (Applied Biosystems) using comparative CT. qPCR primers are listed in *Supplementary file 1a*. mRNA levels from different samples were normalized to a house-keeping gene (*rpsL*) and mRNA levels in CRISPRi strains were compared to that of a vector-bearing Lp02(*dcas9*) control strain using the ΔΔCT method (*Schmittgen and Livak, 2008*) to determine fold repression.

### CRISPRi intracellular growth assays

U937 monocytes (ATCC CRL-1593.2) were maintained in DMEM (Mediatech 15-013-CV)+10% FBS+glutamine at 37°C. Mycoplasma was monitored in the U937 macrophages using the Venor GeM Mycoplasma Detection Kit (Sigma-Aldrich #MP0025). Two days prior to challenge with *L. pneumophila*, cells were plated on 24-well plates at $3\times10^5$ cells/well with 0.1 µg/mL 12-*O*-tetradecanoylphorbol-13-acetate (TPA, Sigma-Aldrich P1585) to promote differentiation. *L. pneumophila* strains bearing each MC array were incubated under inducing conditions, as described above, and added to differentiated U937 macrophages in DMEM+10% FBS+glutamine containing 40 ng/mL aTC at a multiplicity of infection (MOI) of 0.05. Plates were centrifuged for 5 min at 200 × *g* to increase bacteria-macrophage contact. After a 2 hr incubation at 37°C, extracellular bacteria were removed by washing the macrophages twice with DMEM+FBS+glutamine media containing 40 ng/mL aTC. Both 2 and 72 hr post infection (hpi) intracellular bacteria were collected upon macrophage cell lysis by addition of digitonin (0.02% final concentration), serially diluted, and spotted on CYE plates. Results are given as fold growth (colony forming units [CFUs] harvested 72 hpi *vs.* 2 hpi) compared to that of the vector-bearing Lp02(*dcas9*) control.

*A. castellanii* (ATCC 30234) were maintained in PYG media (*Moffat and Tompkins, 1992*) at 25°C. The day before challenge with *L. pneumophila*, cells were plated on 24-well plates at $3\times10^5$ cells/well. The morning of challenge, the media in the *A. castellanii* plates was changed from PYG to AC buffer (*Moffat and Tompkins, 1992*) to promote starvation for 2 hr at 25°C. *L. pneumophila* strains bearing

each MC array were incubated under inducing conditions, as described above, and added to the starved *A. castellanii* in AC buffer containing 40 ng/mL aTC at an MOI of 0.03. Plates were centrifuged for 5 min at 200 × *g* to increase bacteria-amoeba contact. After a 2 hr incubation at 37°C, extracellular bacteria were removed by washing amoeba twice with AC buffer containing 40 ng/mL aTC. Both 2 and 48 hpi intracellular bacteria were collected upon amoeba lysis by addition of saponin (0.05% final concentration), serially diluted, and spotted on CYE plates. Results are given as fold growth (CFUs harvested 48 hpi *vs*. 2 hpi) compared to that of the vector-bearing Lp02(*dcas9*) control.

## Multiplex random CRISPR array self-assembly protocol

R-S-R oligonucleotides were designed as the building blocks for array assembly. For each gene, the reverse complement of the 24 nucleotides downstream of the first PAM sequence (NGG) after the transcription start site was identified and would become the spacer sequence. This sequence was checked to be null of cut sites for SacI (GAGCTC), NotI-HF (GCGGCCGC), and AflII (CTTAAG) as these are used for downstream steps; if not, the sequence adjacent to the subsequent PAM sequence was checked. Spacer sequences were then flanked on either side with one half of the repeat according to the following scheme: top oligonucleotide (5′ to 3′): tgaatggtcccaaaac-24 nt spacer-gttttagagctatgctgttt. Bottom oligonucleotide (5′ to 3′): ttcaaaacagcatagctctaaaac-24 nt complement-gttttgggacca. Standard oligos were ordered from Eurofins Genomics already resuspended in water at 100 µM concentration. Dead end oligonucleotide pairs for R-attB4r and R-attB3r did not contain a spacer but did contain the upstream or downstream repeat sequence fused to the attB4r or attB3r Invitrogen Gateway cloning sequence, respectively.

For array assembly, top and bottom oligonucleotides for a single R-S-R or dead end oligonucleotide pairs were combined 1 µL:1 µL and were phosphorylated in a 50 µL reaction using T4 Polynucleotide Kinase (NEB M0201L) for 2 hr at 37°C. The reaction was stopped by addition of NaCl to a final concentration of 50 mM. Next, 2 µL of each phosphorylated R-S-R oligonucleotide pair for all 44 gene targets and 10 µL of the dead end oligonucleotides were combined in a single microcentrifuge tube. The microcentrifuge tube was placed in a heat block at 95°C for 5 min and then the heat block was turned off and allowed to cool to room temperature for 2 hr. Assembled arrays were preserved through ligation with T4 DNA ligase and its corresponding buffer (NEB M0202L) for 1.5 hr at room temperature. The reaction was purified using a NucleoSpin PCR Clean-up Kit (Macherey-Nagel 740609.250) and eluted in 30 µL 1× TE buffer.

To size-select arrays of interest, four 6.5 µL aliquots of the purified arrays were moved into the interim pDonor4r-3r in parallel by the standard Invitrogen Gateway BP reaction recipe (Thermo Fisher 11789-020) which was allowed to remain at room temperature for 72 hr prior to stopping the reactions with Proteinase K and transformation of all material into *Escherichia coli* GC5 (Genesee Scientific 42-650) with selection on LB+Kan (30 µg/mL) plates. All colonies were scraped off plates using an inoculation loop and added directly to the A1 buffer of a Nucleospin Plasmid Purification Kit (Macherey-Nagel 740499.250) for plasmid isolation and elution in water. A NotI-HF (NEB R3189S) and AflII (NEB R0520S) double digest was performed on the four now recombined eluates, as well as one aliquot of the circular empty pDonor4r-3r plasmid, at 37°C overnight. Next, a 1% agarose gel was used to separate the vector backbone from the excised size-ordered arrays, and we collected bands containing arrays ≥2 plex at 550–650 bps and 650–800 bps to be purified separately using a NucleoSpin Gel Purification Kit (Macherey-Nagel 740609.250). We also collected the cut vector backbone, but only from the empty pDonor4r-3r sample. For all steps forward, the 550–650 bps and 650–800 bps samples were kept separate, but procedures were carried out in parallel. As such, purified arrays were ligated back into the pDonor4r-3r backbone using T4 DNA ligase for 2.5 hr at room temperature, all material was transformed with the plasmid into *E. coli* GC5, and re-circulated plasmids were selected for on LB+Kan plates. All colonies were scraped off plates using an inoculation loop and added directly to the A1 buffer of a Nucleospin Plasmid Purification Kit (Macherey-Nagel 740499.250) for plasmid isolation and elution in 20 µL 1× TE buffer. At this point, individual plasmids can be sequenced using a primer with the sequence: GTTTTCCCAGTCACGAC, if individually purified from a single colony, to confirm successful array assembly.

A Multisite Gateway Pro kit (Invitrogen 12537-100) was used to move arrays into the final vector. To generate the promoter-bearing donor plasmid pMME2162, the *tet* promoter was amplified from pMME1996 with primers containing attB5/attB4 ends for recombination into the pDonorP5-P4 via

a standard Invitrogen Gateway BP reaction. To generate the terminator-bearing donor plasmid pMME2163, the *rrnB* T1 terminator was amplified from pMME1996 with primers containing attB3/attB2 ends for recombination into the pDonorP3-P2 via a standard Invitrogen Gateway BP reaction. Next, the size-selected arrays in the pDonor4r-3r plasmid were combined with pMME985 (tracrDNA in pDonorP1-P5r, described previously; *Ellis et al., 2021*), pMME2162, and pMME2163, and were added to the pMME977 destination vector (attR1/attR2, +*thyA*) by using three times the standard Invitrogen LR reaction recipe (Thermo Fisher 11791-020), with addition of β-mercaptoethanol to a final concentration of 1 mM. The LR reactions were allowed to remain at room temperature for 72 hr prior to stopping the reactions with Proteinase K and electroporation into *E. coli* GC5 with selection on LB+Amp (100 µg/mL) plates. Next, all colonies (generally thousands for 550–650 bps samples and hundreds for 650–800 bps samples) were scraped off plates using an inoculation loop and added directly to the A1 buffer of a Nucleospin Plasmid Purification Kit (Macherey-Nagel 740499.250) for plasmid isolation and elution in 20 µL water. At this point, individual plasmids can be sequenced using a primer with the sequence: CAACCACTTTGTACAAGAAAGCTGGG, if individually purified from a single colony, to confirm successful construct assembly.

Lastly, 10 µL of these final plasmids were introduced into Lp02(*dcas9*) by electroporation and recipient cells were selected for on CYE plates. Colonies were scraped off plates using an inoculation loop, added to 15 mL AYE+Fe+Cys, vortexed, and frozen in 1 mL aliquots with glycerol (final concentration 15%) to be stored at –80°C. Freezer stocks of Lp02(*dcas9*) bearing each of the size-selected portions of a given library (Libraries 1 and 2) are listed in *Supplementary file 1j*. All strains are available upon request.

## MuRCiS intracellular growth assays

*L. pneumophila* bearing each size-selected portion of either Library 1 or 2 were patched from –80°C freezer stocks onto CYE plates. Two days prior to infection *L. pneumophila* cultures of each were grown overnight in 3 mL AYE+Fe+Cys under non-inducing conditions. On the second day, cultures were sub-cultured twice first in the AM to OD600 0.2 with 3 mL fresh AYE+Fe+Cys containing 40 ng/mL aTC and then ~6–7 hr later, to OD600 0.2 with 4 mL fresh AYE+Fe+ Cys containing 40 ng/mL aTC (two cultures each). On the third day, cultures pertaining to either Library 1 or 2 were combined in one tube to serve as a single culture for infection (input vector pool). For Library 1, two different 550–650 bps portions and one 650–800 bps portion were combined. For Library 2, one 550–650 bps portion and one 650–800 bps portion were combined.

U937 monocytes were maintained in DMEM+10% FBS+glutamine and allowed to differentiate into macrophages with 0.1 µg/mL TPA for 3 days prior to *L. pneumophila* challenge. *A. castellanii* were maintained in PYG and starved in AC buffer 2 hr prior to challenge with *L. pneumophila*. Ultimately, differentiated U937 macrophages or starved *A. castellanii* were plated in two to three 10 cm dishes at a density of $1 \times 10^7$ cells per dish.

Infections were carried out with 40 ng/mL aTC in the host cell media at an MOI of 0.05 for differentiated U937 macrophages and 0.03 for starved *A. castellanii*. After addition of bacteria, the 10 cm dishes were spun down at 200 × *g* to increase bacteria-host cell contact and incubated for 2.5 hr at 37°C. Next, host cells were washed in their corresponding media with 40 ng/mL aTC to remove extracellular bacteria and then placed back at 37°C for the duration of the infection, 72 hr or 48 hr for U937 macrophage and *A. castellanii* infections, respectively. To collect input vector pools, bacteria from the cultures used for infection were immediately pelleted by centrifugation and resuspended in 8 mL of RES-EF of the NucleoBond Xtra Midi EF kit prior to proceeding with the standard kit protocol (Macherey-Nagel 740420.50). Precipitated vectors were resuspended in 120 µL of water.

Following the infection duration, *L. pneumophila* were harvested from either the U937 macrophages or *A. castellanii* 10 cm dishes (15 mL vol) by addition of 300 µL of 1% digitonin or 600 µL of 1% saponin, respectively. Next, 10 cm dishes were incubated for 10 min at 37°C, followed by pipette agitation of the cells and collection of the media in 50 mL conical tubes. The conical tubes were shaken vigorously to promote host cell lysis and the *L. pneumophila* were pelleted by centrifugation for 10 min at 3500 × *g*. The resulting bacteria pellets were resuspended in a minimal amount of supernatant and plated on CYE plates. All colonies were scraped off plates using an inoculation loop and resuspended in 8 mL of RES-EF of the NucleoBond Xtra Midi EF kit prior to proceeding with the

standard kit protocol for collection of the output vector pools. Precipitated vectors were resuspended in 120 µL of water.

## PacBio long-read sequencing

Purified vector populations from input and output samples were linearized by addition of 24 µL of SacI restriction enzyme (NEB R0156L) with 16 µL of NEB r1.1 buffer and overnight incubation at 37°C. To purify the linearized vectors, the AMPureXP Bead protocol (Beckman Coulter A63880) was followed using 100 µL of bead slurry. Final eluent volumes varied between samples based on the ability to get a bead-free eluent. Generally, ~10 µL of water was needed for input sample elution and ~40 µL of water was needed for output sample elution. Input DNA concentrations were generally ~200–300 ng/µL and output DNA concentrations were usually 300–700 ng/µL. Vector linearization was confirmed on a 1% agarose gel.

PacBio long-read sequence of the linearized vectors was performed by the NIH NICHD Molecular Genomics Core. Samples were prepared for sequencing on a Sequel instrument (PacBio) using the Sequencing Primer v4 and the Sequel Binding Kit 3.0. Samples were multiplexed and run on multiple SMRT cells simultaneously (# SMRT cells = # multiplex samples divided by 2 or 3) with 10 hr of sequencing camera time.

## Bioinformatics pipeline

PacBio sequencing files (ccs_bam files) were processed using a custom bioinformatics pipeline (https://github.com/GLBRC/MuRCiS_pipeline, copy archived at *Myers and Donohue, 2023*). Total read number and lengths were determined using Samtools (*Li et al., 2009*) (version 1.9) and standard Linux commands (see GitHub for specific commands). All spacer and repeat forward and reverse orientation sequence combinations were used to identify the unique array sequence in the sequencing files using the 'count_spacers_NIH.py' custom Python script (https://github.com/GLBRC/MuRCiS_pipeline, copy archived at *Myers and Donohue, 2023*). Reads lacking an exact match to the repeat sequence were removed from further analysis. Counts for unique spacers were normalized by dividing the read count plus a pseudo-count of 1 by the total number of reads with repeat sequences multiplied by 10,000 in a modified counts per million equation. The 10,000 value was used instead of 1 million due to the lower number of overall reads from this PacBio sequencing.

$$\frac{(Read\ Count + Pseudocount) * 10,000}{(Total\ reads\ with\ repeat\ sequences)}$$

To determine the pairwise correlation of two spacers and plot Chord diagrams, the 'chord_correlation_plot_script.R' script was used. All plots were generated in R (version 4.1.0) and all scripts and command descriptions are available on GitHub (https://github.com/GLBRC/MuRCiS_pipeline, copy archived at *Myers and Donohue, 2023*).

## Deletion strain construction and whole genome sequencing

*L. pneumophila* gene deletion strains were constructed using gene-specific versions of the pSR47s suicide plasmid listed in *Supplementary file 1j* and described recently (*Wibawa et al., 2022*). In our hands, each pSR47s plasmid was introduced into the *L. pneumophila* strain background of choice with the help of pRK600 by tripartite mating as described previously (*Lee and Machner, 2018*). Successful strains were confirmed by failure to grow on CYET+Kan, colony PCR, and whole genome sequencing.

For whole genome sequencing, genomic DNA of *L. pneumophila* was extracted using the Wizard Genomic DNA Purification Kit (Promega A1120), following the standard Gram-negative bacteria protocol, and resuspended in 10 mM Tris-HCl, 1 mM EDTA buffer. The Illumina Nextera XT DNA Library Preparation Kit (#FC-131-1096) was used to prepare libraries for sequencing on an Illumina MiSeq System by the NIH NICHD Molecular Genomics Core. Nucleotide variants were called using HaplotypeCaller (version GATK3, Broad Institute). Integrative Genomics Viewer (version 2.4.14, Broad Institute) was used to visually confirm chromosome deletions. Final deletion strains are listed in *Supplementary file 1j*.

## Deletion strain intracellular growth assays

Intracellular growth assays with deletion strains were performed exactly as for the 10-plex CRISPRi intracellular growth assays except that U937 monocytes were allowed to differentiate for 3 days prior

to *L. pneumophila* challenge and *L. pneumophila* cultures were simply grown overnight without the inducer aTC. To overcome the thymidine auxotrophy of Lp02 strains, thymidine at a final concentration of 0.1 mg/mL was added directly to *L. pneumophila* cultures and to the DMEM+10% FBS+glutamine media to promote growth in U937 macrophages with CFU plating on CYET plates, whereas pMME2400 (+*thyA*) was added to *L. pneumophila* strains by electroporation before growth in *A. castellanii* with selection on CYE plates. For U937 macrophage experiments the control strain was Lp02. For *A. castellanii* experiments that control strain was MML854 (Lp02+pMME2400, G->A mutation at chromosome position 2134550).

## Acknowledgements

We thank the members of the Machner lab for thoughtful scientific discussion during the development of the MuRCiS workflow. We thank members of the NIH NICHD Molecular Genomics Core, especially Tianwei Li and James Iben, for executing the Illumina whole genome sequencing, PacBio long-read sequencing, and for performing initial processing and transfer of data files. Funding: This work was funded by the Intramural Research Program of the National Institutes of Health, USA under Project Number 1ZIAHD008893-12 (to MPM) and the Great Lakes Bioenergy Research Center, US Department of Energy, Office of Science, Office of Biological and Environmental Research under Award Number DE-SC0018409 (to TJD).

## Additional information

### Funding

| Funder | Grant reference number | Author |
|---|---|---|
| Eunice Kennedy Shriver National Institute of Child Health and Human Development | 1ZIAHD008893-13 | Matthias P Machner |
| Great Lakes Bioenergy Research Center | DE-SC0018409 | Timothy J Donohue |

The funders had no role in study design, data collection and interpretation, or the decision to submit the work for publication.

### Author contributions

Nicole A Ellis, Supervision, Conceptualization, Data curation, Formal analysis, Writing - original draft; Kevin S Myers, Writing - review and editing, Formal analysis; Jessica Tung, Anne Davidson Ward, Kathryn Johnston, Katherine E Bonnington, Conceptualization; Timothy J Donohue, Software; Matthias P Machner, Software, Formal analysis, Writing - original draft

### Author ORCIDs

Nicole A Ellis http://orcid.org/0000-0001-9592-8415
Kevin S Myers http://orcid.org/0000-0003-3302-3877
Timothy J Donohue http://orcid.org/0000-0001-8738-2467
Matthias P Machner http://orcid.org/0000-0002-6971-7451

Reviewer #1 (Public Review): https://doi.org/10.7554/eLife.86903.3.sa1
Reviewer #2 (Public Review): https://doi.org/10.7554/eLife.86903.3.sa2
Author Response https://doi.org/10.7554/eLife.86903.3.sa3

## Additional files

### Supplementary files

• Supplementary file 1. Supplementary tables for construct design and gene combinations discovered by CRISPRi. (**a**) 10-plex construct descriptions and list of spacer sequences used in

this study. (**b**) List of transmembrane effector (TME) genes and corresponding spacer sequences targeting them. (**c**) PacBio long-read sequencing metrics. (**d**) Read counts and calculations for all constructs tested in the experiments. (**e**) Subset data for all hits identified in U937 macrophage infection experiments. (**f**) Subset data for all critical gene combinations identified in U937 macrophage infection experiments. (**g**) Subset data for all hits identified in *A. castellanii* infection experiments. (**h**) Subset data for all critical gene combinations identified in *A. castellanii* infection experiments. (**i**) Hits found in single-round CRISPRi experiments. (**j**) Strains and plasmids used in this study.

• MDAR checklist

## Data availability

PacBio CCS BAM sequencing files are available at NCBI SRA under BioProject PRJNA902385 (https://www.ncbi.nlm.nih.gov/bioproject/PRJNA902385).

The following dataset was generated:

| Author(s) | Year | Dataset title | Dataset URL | Database and Identifier |
|---|---|---|---|---|
| Ellis NA, Myers KS, Tung J, Davidson Ward A, Johnston K, Bonnington KE, Donohue TJ, Machner MP | 2022 | Deciphering microbial virulence mechanisms during *Legionella pneumophila* infection | https://www.ncbi.nlm.nih.gov/bioproject/PRJNA902385 | NCBI BioProject, PRJNA902385 |

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

## Appendix 1

### In vitro CRISPR array assembly notes

The in vitro assembly of crRNA-encoding arrays is a key accomplishment of our approach. While it seems simple on paper, the actual assay proved to be technically demanding and required various optimization steps. For experimenters interested in applying our technology to their own research question, the following is a list of strategies we also tried.

### Size selection of CRISPR arrays:

To enrich for longer CRISPR arrays, we explored use of size-selection SPRI beads (Beckman), Pippin Prep instrumentation (Sage Science), and immediate purification from the R-S-R assembly mix by DNA gel electrophoresis. The goal was to maintain the most ligated material and give preference to longer arrays that are outcompeted by smaller arrays during vector incorporation. While these approaches were able to size-select for longer arrays, the highest yield of size-selected arrays occurred when the ligated arrays were first introduced into an interim plasmid, then excised via restriction enzyme digest, size-ordered by gel electrophoresis, extracted from the gel, and then ligated back into the donor plasmid.

### Addition of promoter and terminator:

Attempts to add dead ends bearing the promoter and terminator sequences to the original R-S-R assembly mix led to muddled array assembly which we believed to be the result of the promoter and terminator fragments being much longer (~200 bps) than the R-S-R building blocks (60 bps) Therefore, addition of the promoter and terminator to the final plasmid was accomplished by Invitrogen Multisite Gateway Pro cloning as described.

### Barcoding of arrays:

Barcodes are often used to distinguish between different constructs in library-based experiments. The pooled nature of our de novo array self-assembly protocol thwarts unique barcode addition without interim isolation of each array followed by long-read sequencing to assign each array to a unique bar code. Since long-read sequencing by PacBio sequences each array in its entirety, our MuRCiS pipeline negates the need for barcodes altogether.

### Distal annealing sites:

During the development of our MuRCiS workflow, we also made an array library in which we intentionally designed spacers to encode crRNAs that would target sequences adjacent PAMs further downstream (distal) of the transcriptional start site. We hoped they would serve as good off-targeting controls. However, upon performing the infection experiment, we found almost none of the hits that originally emerged while using the spacers downstream of PAMs most proximal to the transcription start site (*Figures 5 and 6*) and so we would recommend only using these.

