## [Editor Report · eLife assessment]

This **important** study uses CRISPRi to silence multiple effectors in the pathogen, *Legionella pneumophila*. It provides a technique that will allow researchers to address functional redundancy amongst effectors, a problem that has persisted even after decades of study. The methodology used is **convincing**, and further improvement (such as using multiple guides per gene) can lead to the identification of novel virulence factors.

---

## [Referee Report · Reviewer #1 (Public Review)]

The article "A randomized multiplex CRISPRi-Seq approach for the identification of critical combinations of genes" describes the development of a multiplex randomized CRISPRi screening method that they named MurCiS and applied it to study redundancy of L. pneumophila virulence factors. The authors used a L. pneumophila strain carrying dCas9 on the chromosome that they had constructed for a CRISPRi screen they had published recently and here combined it with self-assembly randomized multiplex CRISPR arrays that they developed. The strains carrying the dCas9 and the different CRISPRi arrays were used to infect U937 or Acanthamoeba castellanii cells and the intracellular growth phenotypes were recorded as readout. This allowed the authors to identify certain gene combinations that when knocked down induced a growth defect in either or both cells tested but not when they were knocked down alone. A particular gene combination caught their attention, as the genes lpg2888 and lpg3000 were inducing a growth defect only when both were knocked down in U937 cells but in A. castellanii cells lpg3000 alone was sufficient to cause a growth defect.

The concept of using CRISPRi to look at functional redundancy in effectors is a very useful one to the Legionella field and where biological redundancy limits studies. It has the potential to uncover virulence effectors of importance that have not been described before.

Comments on revised version: In this revised version the authors have answered our concerns satisfactorily except the point related to the use of only one guide per gene.

---

## [Referee Report · Reviewer #2 (Public Review)]

The study by Ellis et al. documents the development of a CRISPR interference (CRISPRi) screen aiming at identifying virulence-critical genes of Legionella pneumophila, the facultative intracellular bacterium causing Legionnaires' disease. L. pneumophila employs the Dot/Icm type IV secretion system to translocate more than 300 different "effector proteins" into host cells. Many effector proteins appear to have redundant functions, and therefore, depleting several of them is required to observe a strong intracellular replication phenotype. In the current study, Ellis et al. develop a "multiplex, randomized CRISPRi sequencing" (MuRCiS) approach to silence several effector genes simultaneously and randomly, thereby possibly causing synthetic lethality for L. pneumophila upon infection of host cells.

The MuRCiS approach comprises the ligation of different CRISPR spacers flanked by repeats in presence of "dead end" oligonucleotide pairs capping a random array of building blocks to be inserted into a library vector. Thus, spacer arrays with an average of 3.3 spacers per array were obtained. As a proof-of-concept, spacer arrays targeting 44 transmembrane effector-encoding L. pneumophila genes were employed to screen for intracellular growth defects in macrophages and amoeba. Consequently, novel pairs of synergistically functioning effector genes were identified by comparative next-generation sequencing of the input and output pools of spacer arrays.

A major strength of this well-written and straightforward study is the construction and use of random and multiplexed CRISPRi arrays, allowing an unbiased and comprehensive screen for multiple genes affecting the intracellular growth of L. pneumophila. The ingenious approach established by Ellis et al. will be useful for further genetic analysis of L. pneumophila infection and might also be adopted for other pathogens employing a large set of (functionally redundant) virulence factors.

The reviewer's suggestion to test the single and double L. pneumophila effector mutant strains for growth in protozoa other than A. castellanii was considered beyond the scope of the current manuscript describing the optimization of the MuRCiS platform. The authors have satisfactorily addressed the minor points raised previously.

---

## [Author Response]

The following is the authors’ response to the original reviews.

We thank the reviewers for recognizing the importance of our work and for their insightfulsuggestions. A point-by-point response to their comments is listed underneath each reviewer’ssection.

**Reviewer #1 (Recommendations For The Authors):**
Major comments1. Have the authors optimized the expression level of dCas9? I cannot find this information in this paper or in their 2021 paper. It is important to avoid the toxicity phenomenon that occurs when using guide RNAs that share specific five base seed sequences (referred to as 'bad seeds').Cui L., Vigouroux A., Rousset F., Varet H., Khanna V., Bikard D. A CRISPRi screen in *E. coli* reveals sequence-specific toxicity of dCas9. Nat. Commun. 2018; 9:1912.Rostain W., Grebert T., Vyhovskyi D., Thiel Pizarro P., Tshinsele-Van Bellingen G., Cui1 L., Bikard D. Cas9 off-target binding to the promoter of bacterial genes leads to silencing and toxicity. Nucleic Acids Research, 2023, gkad170.2. One guide per gene is highly unusual given that different guides block the RNA polymerase with different efficiency. This was even shown by the Machner lab in the Legionella context in Figure 1c of Ellis et al. 2021 for sidM and vipD. Typically, genes need three guides minimum to ensure that the gene of interest is knocked down fully unless it is not possible as the gene is too small and/or it is difficult to find an NGG sequence. The authors have used one guide per effector, how can they be sure that each gene is knocked down? The Machner lab themselves in Figure 3c of Ellis et al. 2021 shows not all genes targeted using multiplex CRISPRi are equally efficiently knocked down. Please justify why only one guide per gene was chosen and add controls to validate the results. The authors themselves state that the strategy of one guide may be problematic. Lines 315-316 it reads... A possible explanation was the incomplete knockdown of a seemingly important process.3. Given what the Machner lab observed about spacer location in Ellis et al. 2021 would it not make more sense to take one set of redundant effectors and make multiplex randomized CRISPRi with them in different locations? For Figure 1 at least.4. Following infection, it seems that the bacteria were not plated onto antibiotic media, so it is not known how well the plasmid harboring guides is kept through infection.Specific commentsA) The first results paragraph describes the set-up of 10-plex synthesized CRISPR arrays, where 10 effector encoding genes of specific gene families are selected. The rationale of the choice of these genes is not given. Please explain.B) Please also add some biological data on what these genes code for, and what is their known or predicted function. It is not very informative and exciting to have tables of lpg numbers without any knowledge of what these genes code for and why they were selected, at least some.C) Figure 1 A Why are only some of the MC arrays shown? Please, at least include in supplementary the others. Again one array in detail would be more informative, showing true knockdown of all genes by qPCR and ideally by western blot.D) I am not convinced that the gene silencing efficiency qPCR comparison is done in the correct way. In my opinion, each of the genes to be knocked down should be tested against the expression of a control gene e.g. rpoS and then these results should be compared and not the results of empty plasmid or CRISPR array containing plasmid directly. L. pneumophila are very sensitive to growth conditions and inoculum, thus the two strains might not be completely at the same growth stage when being compared which can impact the results.E) Figure 1 B As stated in general comment number 4, the authors do not appear to plate onto antibiotic so we don't know how well the plasmid harboring the guides is kept through infection. The sustained presence of the guide is particularly important for CRISPRi.F) The authors found only a few growth phenotypes and mainly this was due to single genes and not combinations of genes. This might again be due to the fact that only one guide per gene was used. How do the authors know that all genes targeted were indeed knocked down?G) Line 119 Alternatively, the genes were not 100% all knocked down, escaping the knockdown effect expected. Could authors take three genes with three guides each and look at impact instead of only one?H) The authors then develop the randomized multiplexed arrays and chose to test 44 TME encoding genes. Line 141 Justify why these effectors were chosen in the text.I) Unfortunately, the method is not clearly described, and many parts are complicated and the text needs to be re-read several times to be understood (lines 150 - 166). Please re-write to better explain to the reader. In line 156 the authors point to a supplementary note 1. This information should be in the main text.J) What is the copy number of the CRISPR plasmid? Please add in the Material and Method section also the origin of this plasmid.Figure 2K) In the paper (line 154-160) and the extra notes, it states that authors attempt to size select CRISPR arrays. However, this is not apparent in Figure 2 schematic. Or are the authors stating that plasmids only containing one guide were selected out? However, line 312 would suggest not. Please clarifyL) A limiting factor in making multiplex guide CRISPR, as the authors are trying to establish in this study, is cloning of multiple guides. In the pre-determined CRISPR arrays in this study, the guides were synthesized. For the randomized multiplex CRISPR in this study, the authors adapt a Golden Gate cloning method to generate multiple sgRNAs in the Cas9 vector. A similar protocol was established in the below paper. Please add this reference.Zuckermann, M.; Hlevnjak, M.; Yazdanparast, H.; Zapatka, M.; Jones, D.T.W.; Lichter, P.; Gronych, J. A novel cloning strategy for one-step assembly of multiplex CRISPR vectors. Sci. Rep. 2018M) As the authors note, Zuckermann et al. similarly note that plex of 3 or 4 is most common and above 5 is rare. This thus appears to still be the limiting step of multiplex CRISPR technology. Please discussFigure 4N) The idea of multiplexed CRISPRi seq to address the biological phenomenon of redundancy is an interesting one, however, I am missing the in-depth functional characterization and discussion of at least one of the redundant functions discovered. Please add.Figure5/6O) As noted above, the strength of the experiments is undermined by how CRISPRi is set up. Having an average multiplex of 2 or three and again only using one guide per gene weakens the study and the results obtained. Furthermore, as stated in general comment number 4, the authors do not appear to plate onto antibiotic so again, we don't know how well the plasmid harboring the guides is kept through infection. The sustained presence of the guide is particularly important for CRISPRi. Please add a validation that the guides are all present.

Response to Reviewer #1

We are grateful to the reviewers for their insightful comments and suggestions on how to further improve the manuscript.

Regarding the issue of ‘bad seed sequences’ (comment #1), we had previously evaluated the expression level of dcas9 (plotted in Figure 1b of the 2021 Communications Biol paper) and tuned our induction conditions accordingly (40 ng/mL as described in the Methods). Since all strains used in this study express dcas9 from the chromosome, not a plasmid, this eliminates the possibility of fluctuations in expression levels due to variabilities in plasmid copy numbers.

In the rare event that toxicity by any given guide occurs, we would expect that guide to already be underrepresented or missing in the input pool following 24+ hours of CRISPRi induction during axenic growth. Our data, now discussed in the manuscript (Lines 211-216 and Figure S2), revealed that this was not the case and that all guide-encoding spacers were present in roughly equal amounts (median of >5000 occurrences). As with any knockdown study, the creation of true chromosome deletions was performed throughout as to alleviate the issue of false positives.

Regarding comments #2, #3, and specific comments made under point F, G, and O, on the topic of using single vs. multiple guides, we agree that there are circumstances under which using more than one guide per target may be advantageous, for example when attempting to delete a gene from mammalian cells using conventional CRISPR engineering. In the study described here, this is not the case. In fact, we did create a second array library with alternative guides targeting the same group of genes at locations other than the “optimal location” identified in our 2021 paper and found that these “sub-optimal” guides were inefficient for identifying critical effectors as described in Supplemental Note S1 under the heading “Sub-optimal annealing sites” (Lines 919+). These data suggest that adding sub-optimal guides to the arrays of optimal guides might ‘poison’ the arrays and limit rather than enhance their ability to identify gene combinations.

Regarding comment #2, #3, and specific comments made under point C, F, and G, on the topic of confirming efficient gene knockdown for the identification of critical genes, we remind Reviewer 1 that we did confirm knockdown of 60 of the target genes of the 10-plex screen to be at least 2-fold, with an average fold repression of one order of magnitude or more (Figure 1A). While knockdown of every gene in every 10-plex construct would be an unprecedented ask of any published CRISPR screen, we believe that these 60 genes provide a large enough sampling of all guides to elucidate the range of knockdown to be expected by our CRISPRi platform. As with other knockdown technologies, such as RNAi, there is no expectation of accomplishing complete knockdown for any given target. Hence, the data in Figure 1A suggest that the lack of identifying critical genes using pre-determined 10-plex arrays was not due to a lack of knockdown efficiency, but rather the difficulty to accurately predict redundancy within a cohort of uncharacterized genes, accentuating the need for array randomization with MuRCiS.

On the topic of antibiotic use for plasmid selection (comments #4, E and O), we would like to clarify that the CRISPR plasmids were selected by thymidine prototrophy, not antibiotic resistance, and we apologize for not making this clearer. The laboratory strain Lp02 is a thymidine auxotroph (thyA-) L. pneumophila variant, and plasmid retention is routinely achieved by including the thymidine biosynthesis gene (thyA) on the plasmid backbone. Only with a plasmid bearing the thyA gene can L. pneumophila grow on CYE (thymidine-) plates. Our use of vectors bearing thyA and plating on CYE plates is described in the Methods section. Further, in Figure 7 of our 2021 paper, we show that CRISPR plasmids are efficiently retained by Lp02 for the duration of a 48-hour infection, resulting in efficient multi-gene knockdown even at the end of the intracellular growth experiment.

Regarding comments A and B, on publishing the biological data used to classify genes in gene families for 10-plex silencing, we do not consider it critical to provide additional information beyond the broad classification (e.g. kinases, phosphatases, etc) described in Table S1. Structural predictions constantly change due to continuously evolving databases. Our initial analyses were made in 2015 using HHPRED Hidden-Markov models and, in all likelihood, those predictions have been refined since then. Moreover, with the recent advent of Alphafold, anyone interested in learning more about select effectors from our list is advised to simply access the most recent functional predictions directly on the Alphafold webpage (https://alphafold.ebi.ac.uk/). We clarify how predictions were made on Lines 97-101.

Regarding specific comment D, on our method for qPCR normalization and comparison, we point Reviewer 1 to the Methods section (Lines 460+) where we describe that data obtained from each CRISPRi strain were in fact normalized to the levels of rpsL prior to comparing them to the normalized data from the strain with the empty control plasmid. This normalization to rpsL, a gene encoding a ribosomal protein, also corrects for growth differences between samples.

Regarding specific comment H, the justification for studying 44 transmembrane effector-encoding genes was driven by the fact that activities mediated by transmembrane proteins are difficult (though not impossible) to be replaced by cytosolic proteins, for example the transport of metabolites across the LCV membrane. And since transmembrane regions can be predicted with high confidence, we decided to probe this group of TMEs for synthetic lethality with the randomized CRISPRi approach as proof-of-concept. To make this clearer, we have added more detail to the text (Lines 151-155).

Regarding specific comment I, we have further simplified the description of the cloning technique to increase clarity (Lines 156+). The information listed under Supplemental Note S1, though informative, is not critical for the overall understanding of this highly technical section, and since the reviewer already considered this section to be difficult to follow, we would prefer to not further complicate the text by including these non-essential details.

Regarding the origin of the CRISPRi plasmid (specific comment J), we point Reviewer 1 to the reference (Hammer BK and Swanson MS (Mol Microbiol 1999)) listed in Table S10: Strains and Plasmids Used in this Study.

Regarding specific comment K and O, on the clarity of depicting the CRISPR array size selection process, we have updated the Figure 2 schematic. Reviewer 1 is correct in that despite our best efforts to exclude short CRISPR arrays, inevitably some 1-plex arrays remained in our input vector pool. Still, the average length of all arrays used in our pilot study exceeded three crRNA-encoding spacers. Further, having a population of 1- or 2-plex arrays in our libraries did allow us to pin-point the most critical effectors of a larger arrays within the same MuRCiS experiment (Table S5 and Table S7), a strength of MuRCiS as described in the discussion (Lines 378+).

Regarding specific comment L, we appreciate Reviewer 1’s suggestion of an additional reference and we have added it to the manuscript as reference #23 (Line 71). While this reference does use a Golden Gate strategy to build a multiplex array, that array was not randomized but had a predefined order. Hence, our assembly method is unique due to its randomization.

Regarding specific comment M, on array length cloning limitations, we agree with the conclusion of Zuckermann in Figure 1d of their article that longer inserts are generally harder to get into vector backbones. The challenge of cloning longer inserts is a common phenomenon of general biology and is not unique to cloning CRISPR arrays. We have altered the wording in our manuscript to better describe the intrinsic competition between short and long inserts during cloning (Lines 162-164).

Regarding specific comment N, we second Reviewer 1’s desire to learn more about the critical effector pairs discovered here. With that said, the goal of this manuscript is to report the development of a novel MuRCiS pipeline to identify these critical pairs. Biochemical and molecular investigations of the encoded protein pairs are on-going and will be the topic of a future manuscript.

**Reviewer #2 (Recommendations For The Authors):**
Specific points1. The effector repertoire of L. pneumophila seems to have evolved in response to the plethora of potential protozoan hosts (PMID: 31988381). To further assess evolutionary aspects of the vast L. pneumophila effector arsenal, it would be interesting to test the single and double effector mutant strains (Fig. 5FG, Fig. 6EF) for growth in protozoa other than A. castellanii.1. Most CRISPR arrays comprising genes encoding functionally similar proteins or encoding evolutionarily conserved proteins did not substantially affect intracellular growth of L. pneumophila (Fig. 1B). This rather surprising result should be further discussed.1. l. 118/119: "Similar results ..., where none of the MC arrays ..." This statement should be phrased more precisely, since some CRISPR arrays did indeed have an effect on intracellular growth of L. pneumophila in U937 macrophages, while none affected intracellular growth in A. castellanii (Fig. 1B).1. Typos:l. 852: ... (arbitrarily set to -100).l. 862: ... Legionella-containing vacuole (LCV).l. 895: ... and so we would recommend ...

Regarding point 1, we thank Reviewer 2 for the suggestion of testing effector mutants in different hosts. While the primary purpose of the current manuscript was to optimize the MuRCiS platform, future studies using this technology to investigate specific biological questions related to Legionella infection would certainly benefit from including more than one amoebaean species.

Regarding point 2, we agree that the lack of substantial growth defects seems surprising. Yet only two of the seven core effectors (found in all Legionella sp.), lpg2300 and mavN, individually attenuated Legionella intracellular growth when deleted (Burstein 2016 Nat Genetics; Isaac et al., 2015 PNAS). Thus, we hypothesize that the functions many effectors fulfil are of such importance for intracellular survival that that redundancy reaches beyond the boundary of conservation or like-function. We have added a statement emphasizing this at the end of the Figure 1 results section (Line 122-125).

Regarding points 3 and 4, we thank Reviewer 2 for catching these errors and have corrected where needed in the text.

-l. 852 (now Line 874): … (arbitrarily set to -100,000) is correct for Figure 6E.